# The Dilemma of HSV-1 Oncolytic Virus Delivery: The Method Choice and Hurdles

**DOI:** 10.3390/ijms24043681

**Published:** 2023-02-12

**Authors:** Guijin Tang, Dawei Wang, Xiangqian Zhao, Zhihua Feng, Qi Chen, Yangkun Shen

**Affiliations:** Fujian Key Laboratory of Innate Immune Biology, Biomedical Research Center of South China, Qishan Campus, Fujian Normal University, College Town, Fuzhou 350117, China

**Keywords:** cancer immunotherapy, HSV-1 oncolytic virus therapy, herpes simplex virus type 1, intravenous injection, mode of administration, oncolytic virus

## Abstract

Oncolytic viruses (OVs) have emerged as effective gene therapy and immunotherapy drugs. As an important gene delivery platform, the integration of exogenous genes into OVs has become a novel path for the advancement of OV therapy, while the herpes simplex virus type 1 (HSV-1) is the most commonly used. However, the current mode of administration of HSV-1 oncolytic virus is mainly based on the tumor in situ injection, which limits the application of such OV drugs to a certain extent. Intravenous administration offers a solution to the systemic distribution of OV drugs but is ambiguous in terms of efficacy and safety. The main reason is the synergistic role of innate and adaptive immunity of the immune system in the response against the HSV-1 oncolytic virus, which is rapidly cleared by the body’s immune system before it reaches the tumor, a process that is accompanied by side effects. This article reviews different administration methods of HSV-1 oncolytic virus in the process of tumor treatment, especially the research progress in intravenous administration. It also discusses immune constraints and solutions of intravenous administration with the intent to provide new insights into HSV-1 delivery for OV therapy.

## 1. Introduction

As a promising non-specific tumor immunotherapy, an oncolytic virus (OV) therapy has been clinically proven to have the potential of specifically lysing tumor cells, causing the release of tumor-specific antigens, and inducing anti-tumor immune responses. Numerous related clinical trials are currently being conducted in a variety of tumor types [1,2,3,4,5,6,7]. In addition, compared with other tumor treatment methods, such as chemotherapy, targeted therapy, and anti-programmed death (PD)-1/PD-ligand 1 (PD-L1) therapy, OV therapy has the advantage of being able to be customized according to clinical requirements using in vitro gene editing, possessing a better ability to lyse tumors and activate immunity, and meeting the demands of cancer patients who require personalized treatment [8].

Currently, commonly used OVs include herpes simplex virus type 1 (HSV-1), adenovirus, poxvirus, Newcastle disease virus, and reovirus. Each type of OV has a specific cell entry mechanism and reacts differently with various tumor cells [9,10]. Among them, HSV-1 is considered to have promising clinical applications owing to its potent lytic ability, the broad spectrum of infected cells, ease of genomic modification, induction of long-term cellular immune responses, and availability of drugs for control of its proliferation [11]. Four OV drugs have so far received clinical approval, two of which are HSV-1 oncolytic virus, including Amgen’s HSV-1-based talimogene laherparepvec (T-VEC), which the U.S. Food and Drug Administration approved in 2015 for the treatment of unresectable metastatic melanoma [12,13] and Daiichi Sankyo Teserpaturev (G47∆), which received clinical approval in Japan in 2021 for the treatment of malignant glioma [14,15]. Notably, in the phase III clinical trial of T-VEC combined with PD-1 anti-tumor drug Keytruda in the treatment of melanoma, the tumor remission rate reached as high as 62%, with 33% complete remission, further setting off a wave of OV combination therapy studies [16].

The administration of HSV-1 oncolytic virus therapy is mainly based on intratumoral injection, and this method has undergone numerous clinical trials [17,18]. Studies have revealed that intratumoral injection can precisely deliver OV to some superficial tumor sites; however, locally generated anti-tumor immunity remains to be improved for the treatment of systemic metastases [19]. Since cancer-related deaths are often attributed to the systemic spread of the tumor, delivery of OV to distant metastatic tumor sites is essential for administering OV therapy. Intravenous administration of OV drugs is a systemic delivery method, which, in principle, can solve the problem of systemic distribution of OVs. However, many obstacles remain as stated in the following: 1. Once OVs enter the blood circulation, the body’s antiviral immune response is activated, preventing OVs from effectively reaching target tumors. The viral load that reaches the tumor site is greatly reduced [20]. 2. There are a large number of macrophages in the reticuloendothelial tissue in the human lung, liver, and spleen, which have a strong ability to clear OVs [21,22]. 3. A large number of OVs enter the blood, which may cause the body to produce a cytokine storm and, in turn, trigger an inflammatory response, such as chills, high fever, and irreversible damage to some normal tissues and organs. 4. OV migrates more easily to the trigeminal nerve through the circulatory system, which could pose a latent risk.

Therefore, the investigation of more efficient modes of OV administration is a research priority in the field of OV therapy. This review will briefly reiterate the mechanism of HSV-1 oncolytic function and present studies in detail on different modes of OV administration, especially intravenous administration, and strategies to improve its effectiveness.

## 2. Mechanisms of HSV-1 Oncolytic Virus Therapy

As an excellent gene delivery platform, HSV-1 oncolytic virus has two major advantages over other gene delivery vectors: replicability and large gene capacity [23]. HSV-1 oncolytic virus infects and lyses tumor cells through immediate early gene inhibition of cell death pathways in the early stages, and subsequently, when the virus continues to proliferate and assemble, cellular autophagy, apoptosis, and necroptosis are prompted to lyse tumor cells [24]. Furthermore, HSV-1 oncolytic virus can cause immunogenic cell death (ICD) in tumor cells, which triggers the production of tumor-associated antigens, pathogen-associated molecular patterns (PAPMs), cellular risk-associated molecular patterns signaling (e.g., heat shock proteins, high mobility group box protein 1, calreticulin, and adenosine 5′-triphosphate), and cytokines [e.g., chemokine (C-X-C motif) ligand 2 (CXCL2), CXCL10 and stimulation of T cell recruitment of chemokines, type I interferons (IFN- I), tumor necrosis factor (TNF)-α, and interleukin (IL)-12)] [25,26]. These released molecules recruit immune cells, such as natural killer cells (NKs) and neutrophils, which enhances tumor antigen presentation and immune responses in the tumor microenvironment (TME), breaks immune tolerance in TME, transforms cold tumors into hot tumors, and promotes efficient antitumor responses [27,28].

In addition, HSV-1-induced ICD of tumor cells promotes antigen-presenting cell (APC) infiltration into the tumor and its maturation. This is followed by the activation of antigen-specific CD4^+^ and CD8^+^ T cell responses in the draining lymph nodes, resulting in the expansion of CD8^+^ T cells into cytotoxic T lymphocytes (CTL), generating the body’s de novo immunity against the tumor. Eventually, an “in situ vaccine” is formed at the tumor site. HSV-1-activated CTLs promote the regression of distant metastases through the circulatory system, producing an “abscopal effect” (Figure 1).

In conclusion, HSV-1 oncolytic virus, a multifunctional tumor immunomodulatory drug, can inhibit angiogenesis, regulate tumor metabolites, or combine with other anticancer drugs to produce antitumor effects [11,29]. However, owing to the lack of clinical data, it is unclear whether these functions directly affect tumors. Therefore, further studies are required.

## 3. Research Progress on Various Administration Modes of HSV-1 Oncolytic Virus in Tumor Therapy

Currently, mainly intratumoral, thoracoabdominal, and intravenous injections have been employed for OV administration. Each delivery method has its characteristics, and the choice of delivery method is crucial for the efficacy of HSV-1 oncolytic virus when treating different types and locations of tumors. The characteristics of different drug delivery methods and related research progress will be discussed in the following section.

### 3.1. Intratumoral Injection Administration

Clinically approved HSV-1 oncolytic virus is administered by intratumoral injection [15,30]. Effective tumor remission was attained in approximately 16.3% of patients in a clinical trial involving 50 patients with malignant melanoma who received intratumoral injections of T-VEC, and those who received T-VEC survived about 20 months longer on average compared with the control group [31]. In another clinical phase II trial, 13 patients with malignant glioma were treated with intratumoral injection of Teserpaturev (G47∆). Following 1 year, the survival rate was 92.3%, as opposed to the conventional drug treatment’s survival rate of only 15% [32,33,34]. In addition, some studies have reported that T-VEC can not only directly infect and lyse tumor cells at the injection site but also activate a distal anti-tumor immune response [35]. Puzanov I’s research revealed that 26 patients (74%) had ≥50% lesion regression in 35 measurable lesions directly injected with T-VEC and 12 (52%) of 23 uninjected measurable lesions had ≥50% lesion regression [19].

However, intratumoral injection of OV is usually recommended for the treatment of superficial tumors, such as sarcoma and melanoma, which are more accessible. The extracellular matrix serves as the primary barrier for viral entry into solid tumors, making intratumoral injection administration challenging. To overcome this problem, gene-editing technology has been employed to induce OVs to express relaxin, thereby selectively degrading the extracellular matrix to facilitate viral entry into tumor cells [36].

### 3.2. Thoracoabdominal Injection Administration

Thoracoabdominal injection administration refers to the injection of OV directly into the cavity, followed by the absorption of the virus through local veins, acting on local tumors or through blood circulation to deliver OV to the tumor lesion sites of specific organs or tissues, i.e., thoracic and intraperitoneal injections. The thoracic injection is mainly used for the treatment of malignant pleural effusion, malignant pleural mesothelioma, and other tumors in the thoracic, whereas the intraperitoneal injection is mainly used for the treatment of gastric cancer, colorectal cancer, and mesothelioma [37,38,39]. A study of liver metastases from colon cancer in *mice* revealed that HSV-1-like oncolytic virus ribonucleotide reductase-defective HSV-1 (hrR3) significantly reduced tumor burden and prolonged survival in *mice* [40,41]. In addition, the study revealed that compared with intravenous injection, intraperitoneal injection caused lower detection of the hrR3 virus in the heart, lung, and kidneys than that of intravenous injection. Additional data suggested that the median survival of hrR3 *mice* injected intraperitoneally was approximately twice that of those injected intravenously and that it was more effective in reducing the tumor burden of the *mice* [42].

The use of thoracic injections of T-VEC to treat malignant pleural effusion caused by tumor metastasis has recently entered phase II clinical trials [43]. In addition, a clinical trial that involves administering HSV1716 into the thoracic cavity via a pleural placement tube for the treatment of malignant pleural mesothelioma has entered clinical phase II trials [44]. However, there is insufficient clinical evidence to suggest that thoracic injection delivery improves the effectiveness of OV therapy.

The use of thoracic injection of OV requires a higher dose and more safety measures than intratumoral injection owing to their long residence time in the cavity. Additionally, thoracic injection of OV requires a pre-buried tube, which is prone to pleural infection. Intraperitoneal OV that is absorbed through the peritoneal vein will pool in the portal vein and run the risk of entering the hepatic system, resulting in severe damage. Therefore, the use of thoracoabdominal injection for the administration of OV drugs should be evaluated carefully.

### 3.3. Intravenous Injection Administration

Currently, intratumoral injection remains the primary method of OV drug delivery; however, owing to the complexity of its surgical procedures and the barriers to OV delivery to distal metastases, intravenous injection of OV, a simpler mode of drug delivery, has been the focus of research and development in the field [30].

#### 3.3.1. Advantages of Intravenous Injection

Intravenous administration of OV is a systematic delivery method in which OV is injected into peripheral veins to migrate to the tumor site via the circulatory system and exert lytic effects [45]. Firstly, the prominent advantage of intravenous injection over local injection is that it can treat advanced systemic metastatic tumors. For example, for astrocytoma, pancreatic cancer, liver cancer, and other tumors, local injection is very difficult, while the intravenous injection is easy to administer and, therefore, it is a favorable choice for the treatment of these diseases. Secondly, owing to the rich vascularity inside the tumor, local injection of OV requires a complicated surgical procedure that increases the risk of infection for patients and increases the possibility of tumor rupture, bleeding, tumor tissue shedding, and metastasis. Thirdly, intravenous injection also facilitates the development of combination therapy. Currently, OV therapy is frequently combined with other tumor treatment modalities, such as PD-1 antibody, an immune checkpoint inhibitor, to achieve better anti-tumor effects, and these therapeutic antibodies are often administered intravenously.

#### 3.3.2. Advances in the Study of Intravenous Injection of Oncolytic Virus

Numerous clinical trials including the intravenous administration of oncolytic poxvirus, Newcastle disease virus, M1 virus, blistering stomatitis virus, and oncolytic adenovirus are currently being conducted [45,46,47]. In the phase 1b study, 67% of patients had radiographically stable disease following intravenous injection of oncolytic vaccinia virus Pexa-Vec [48]. In a phase I trial of a single intravenous infusion of Pexa-Vec in 23 patients with advanced solid tumors, dose-related anti-tumor activity was observed, and normal tissues were not affected clinically [49]. In phase I clinical trial, Pelareorep, a tumor oncolytic ehrlichiosis virus product delivered intravenously, induced a T-cell inflammatory phenotype in pancreatic ductal adenocarcinoma; ehrlichiosis virus replication, T-cell infiltration, and upregulation of PD-L1 were detected in tumor tissue of patients treated with Pelareorep, and cancer was controlled in three of ten patients [47]. Clinical studies on intravenous HSV-1 oncolytic virus are limited [50]. We speculate that this may be related to the characteristics of HSV-1, such as HSV-1 having larger viral particles, more surface proteins, and being removed by the immune system more readily. However, there are still numerous studies actively investigating the possibility of intravenous administration of HSV-1.

Genetically engineered HSV-1 oncolytic virus that is administered intravenously has been shown to be effective against tumors in preclinical animal studies. Luo Y et al. revealed that tumor growth was significantly inhibited by intravenous injection of Ld0-GFP, an HSV-1 oncolytic virus targeting hepatocellular carcinoma (HCC), compared with controls. Additionally, they demonstrated that intravenous Ld0-GFP treatment induced robust tumor eradication and durable cure without recurrence in 62.5% of *mice* implanted with H22 tumors [51]. Braidwood et al. reported that intravenous injection of HSV1716, an ICP34.5 double-copy mutant of knockout HSV-1, was 100% effective in treating a *mice* model of HuH7 HCC [52]. These results suggested that intravenous administration of HSV-1 oncolytic virus has an efficient anti-tumor effect, which provides a basis for clinical trials of liver cancer treatments.

Moreover, several studies of systemic dosing based on hepatic arterial perfusion could provide a reference for the intravenous administration of HSV-1 oncolytic virus. For example, a phase I clinical trial revealed that patients with liver metastases from colorectal cancer who received NV1020 OV transhepatic arterial infusion (deleted ICP0, ICP4, and neurotoxic factor ICP34.5 from HSV-1) reported stable disease in 7 of 12 patients (58%) and partial responses in 2 patients following 28 days of administration. In addition, none of the patients experienced any significant safety problems following the administration of the drug [53]. Another clinical trial conducted in 2010 demonstrated that 50% of patients had stabilized hepatic metastases and 68% of patients had tumor control following NV1020 infusions [54]. Currently, the first clinical trial for intravenous HSV-1 revealed a good safety profile for modified HSV1716. However, the efficacy of OV could not be evaluated owing to the small sample size involved in the trial and the fact that most patients were treated with other drugs in conjunction with OV [55]. In addition, HSV-1 oncolytic virus MVR-T3011 IV, the first genetically modified HSV-1 oncolytic virus to enter clinical trials and be administered by intravenous injection, is currently in phase I clinical trials. Although no definitive results have been published, *mice* testing has shown that the drug is safe [56]. MVR-T3011 IV is modified on the wild-type HSV-1 gene backbone, carrying PD-1 antibody and IL-12 gene with immunomodulatory functions, ensuring that the virus can selectively target tumor cells with strong replication ability but does not proliferate in normal tissues and cells while being safely attenuated.

To ensure that the virus reaches the tumor site following intravenous administration of HSV-1 oncolytic virus is crucial to prevent and minimize clearance of the virus by the host organism. Systemic administration is a concentration-driven process in which only viral doses beyond a threshold can be detected. There are no definitive clinical data on the precise volume of intravenous HSV-1 oncolytic virus that reaches the tumor site. A recent phase I clinical trial revealed that the intravenously administered oncolytic vaccinia virus Pexa-Vec was only detectable from tumor tissue when the viral load exceeded a threshold of 10^9^ infectious units [49]. Producing sufficient OVs to reach the tumor site without compromising safety is a major challenge since the therapeutic efficacy of OVs is closely related to the dose of the virus.

In addition, it is unclear whether HSV-1 oncolytic virus can cross the blood-brain barrier if the intravenous injection was utilized to treat brain tumors owing to the blood-brain physiological barrier that exists in the human body. Notably, Samson et al. demonstrated that the infiltration of CTL into tumors was enhanced by intravenous administration of tumor oncolytic eutherian virus to patients with a brain tumor, suggesting a possible approach to the treatment of brain tumors by intravenous injection of HSV-1 oncolytic virus [57].

## 4. Potential HSV-1 Immune Clearance Problems in Intravenous Administration

The body’s immune defense mechanisms against viruses mainly include antiviral innate immunity and adaptive immunity (Figure 2). Upon intravenous administration, the initial phase of HSV-1 infection activates the body’s innate immune defense system, promoting the secretion of type I interferon (IFN-I). The early antiviral effect is also significantly influenced by the monocytes, neutrophils, dendritic cells (DC), macrophages, and NK cells [58,59,60]. In the late stages of infection, with the activation of adaptive immune responses, antigen-specific T cells and B cells play a key role in the resistance to viral infection, CD8^+^T can control the migration of HSV-1 from the site of infection to the nervous system and inhibit the reactivation of latent virus [61]. B lymphocytes are rapidly activated at the early stage of HSV-1 infection and produce large amounts of anti-HSV-1-IgM [62]. In this report, we will elaborate on the potential immune clearance problems with intravenous administration that are suggested to be induced by HSV-1, and discuss the mechanisms that are known to be, or might be, activated in response to HSV-1 infection.

### 4.1. Antiviral Innate Immunity

#### 4.1.1. Antiviral Effect of IFN-I

The IFN-I signaling pathway is the primary defense mechanism for the host against HSV-1 infection [63]. As depicted in Figure 3, HSV-1 infection activates Toll-like receptors (TLRs), retinoic acid-inducible gene l-like receptors (RLRs), and DNA receptor cyclic GMP-AMP synthase (cGAS) via PAPMs (including viral capsids, nucleic acids, or proteins), which in turn induce the production of IFN-I [64,65,66,67]. Among them, TLR2 [68], TLR3 [69], and TLR9 [70] recognize HSV-1 and activate antiviral responses. IFN-I exerts its antiviral effects in the following main four ways: (1) It directly promotes virally infected cells’ death. IFN-β induces apoptosis by blocking cell cycle progression [71]. (2) It activates several IFN-stimulated genes (*ISGs*) through the Janus kinase/signal transducers and activators of the transcription (JAK-STAT) pathway to produce a comprehensive antiviral effect [72]. (3) It triggers IFN-induced activation of dsRNA-dependent protein kinase R (PKR), which is also an important mechanism to resist viral infection [26,73]. (4) It recruits APCs, promoting their maturation and activating antigen-specific CD4^+^ and CD8^+^ T cell-mediated antiviral immunity [74].

#### 4.1.2. Antiviral Effect of Immune Cell

As a class of immune cells that regulate inflammatory responses, neutrophiles can not only eliminate pathogens by phagocytosis and release of neutrophil extracellular traps (NETs) in antiviral innate immunity but also regulate antiviral immune responses in antiviral adaptive immunity [79,80,81,82]. Macrophages with the M1 and M2 phenotypes can eliminate viruses through cytophagy and phagocytosis, and process and present viral antigens to form complexes with major histocompatibility complex (MHC) II molecules to activate subsequent adaptive antiviral immunity [83,84,85]. NK cells also play a critical antiviral role in the early stage of HSV-1 infection. Activated NK cells can not only kill virus-infected cells by releasing perforin or granzymes but also promote the production of cytokines, such as IFN-γ or TNF-α to attract more immune cells to exert antiviral effects [86]. Furthermore, NK cells play a major role in local HSV-1 viral clearance. Dendritic cells (DCs) interact with the virus and can effectively control its spread [87]. DCs can phagocytose virus-infected cells or HSV-1 and process and present viral antigenic peptides, thereby, activating specific antiviral immunity in CTL [88]. DCs play a regulatory role in the body’s antiviral immunity by recognizing PAPMs of HSV-1 using abundant TLRs on their surface and secreting antiviral-related cytokines, such as IFN-I, IL-6, and IL-12 [89].

### 4.2. Antiviral Adaptive Immunity

CD8^+^ T cells play a central role in host-specific antiviral immunity and viral clearance [90]. Following HSV-1 infection, CD8^+^ T cells interact with APCs such as DCs; APCs present viral antigenic peptides to the cell surface MHC class I, attracting CTLs to migrate to the site of viral infection. CTLs can achieve viral clearance by killing virus-infected cells in various ways [91,92,93,94,95,96]. In addition, CD4^+^ T cells can also effectively regulate the antiviral immune response and enhance host defense against viruses. CD4^+^ T cells can be activated after the binding of viral antigenic peptides delivered by DCs to MHC II molecules. Activated CD4^+^ T cells can provide the relevant costimulatory signals and cytokines for initial CD8^+^ T cell activation and B cell-mediated antibody response [97,98,99,100,101,102,103,104,105]. B cell-mediated specific humoral immunity is important in resistance to HSV-1 infection. Antibodies of the HSV-1 have been observed in 60% of the population. Intravenous HSV-1 oncolytic virus is rapidly neutralized by pre-existing antibodies and complement in patients who have been infected and still have antibodies to the HSV-1 in their bodies [62,106].

In summary, the body’s immune system is effective in eliminating HSV-1. However, it is challenging to adopt a single approach to counteract the body’s comprehensive immunity during the intravenous administration of OV. Therefore, it is necessary to consider the main immune factors of viral clearance, which are important to the systemic administration of OV.

## 5. Strategies to Overcome Immune Clearance and Enhance Targeting of Tumors

### 5.1. Overcoming Immune Clearance

The following strategies have been proposed or used to overcome immune clearance for intravenous OV application:

#### 5.1.1. Combined with Immune Inhibitor

Intravenous administration of OV combined with natural immune pathway inhibitors can reduce the virus clearance by the natural immune system in peripheral blood [107,108]. For example, the immunosuppressive agent cyclophosphamide has been used to improve the systemic delivery of oncolytic HSV-1 in the treatment of tumors [109]. Cyclophosphamide has been demonstrated to increase OV transmission, transgene expression, and antitumor efficiency by limiting innate immune responses in animal models [110]. In addition, it can inhibit neutralizing antibody induction, macrophage, regulatory T-cell induction, and IFN-γ production [111,112,113,114,115,116]. Phosphatidylinositol 3-kinase δ (PI3Kδ) is a key regulator of macrophage that enhances intravenous delivery of OV to tumors. Idelalisib, a PI3Kδ inhibitor that protects OV from phagocytosis by macrophages, has also been used in vivo to enhance the anti-tumor efficacy of intravenously delivered OVs, reduce the tumor burden, and prolong survival [117]. Thus, this strategy utilizes a combination of immunosuppressant and OV therapy to facilitate cancer treatment (Figure 4A).

#### 5.1.2. Liposome and Nanopolymer Encapsulation

It has been demonstrated that the use of polymers, liposomes, and nanoparticles encapsulated with HSV-1 can prolong the virus circulation, reduce the uptake of HSV-1 by hepatocytes and Kupffer cells, and preferentially colonize tumor sites following intravenous injection (Figure 4B–D). Surface modification of oncolytic adenovirus using polymers that are either chemically conjugated or physically engineered through electrostatic interaction can help evade antiviral immune responses and ensure viral accumulation at the tumor site [118]. Several studies have demonstrated that covalent conjugation of the virus to polyethylene glycol (PEG) reduces immune response [119,120]. In addition, modification with PEG improves the colloidal stability of the virus and prolongs its circulation time in the blood [121]. Covalent modification of the surface of tumor lytic cells HSV-1 G207 with folate-PEG conjugate (FA-PEG) can reduce the immunogenicity of HSV-1 and enhance specific targeting of tumor cells with folate receptor overexpression. In BALB/c nude *mice* with subcutaneous KB xenografts tumors, the intravenous FA-PEG-HSV group exhibited higher anti-tumor efficiency and tumor targeting specificity [122]. In the presence of neutralizing antibodies, systemic intravascular delivery of hrR3 complex, an OV HSV-1 mutant encased in a liposome, has been reported to be effective in the treatment of numerous liver metastases [123]. Lv et al. used bioengineered cell membrane nanovesicles (BCMNs) carrying targeted ligands to produce BCMN-camouflaged tumor OV (OA@BCMNs). In vitro studies have demonstrated that OA@BCMNs successfully evaded immune clearance and accumulated in tumor tissues, where they could efficiently infect cancer cells and replicate inside them [124]. With the help of these molecules, the OV can avoid direct contact with immune cells in the blood, which causes systemic antiviral reactions and eliminates the OVs.

#### 5.1.3. Cellular Delivery System

Delivery of the virus by various cells is another approach to evade innate immune clearance and enhance tumor targeting. For example (Figure 4E), mesenchymal stem cells (MSCs), neural stem cells, chimeric antigen receptor T (CAR-T) cells, T cell receptor-engineered T cells, and tumor-infiltration lymphocytes have all been used to deliver oncolytic adenovirus and myxoma virus to reduce the immunogenicity of the virus [125,126,127,128,129]. Currently, it is believed that the dynamic interaction between vector cells and the host occurs through vector response to tumor-secreted chemokines. It has been demonstrated that administrating MSCs that have been infected with oncolytic HSV (MSC-oHSV) can effectively track the metastases of tumors and detect the presence of the virus in the brains of brain-tumor-bearing *mice*. The effective rate is approximately twice that of the control group and significantly prolongs the survival of brain-tumor *mice* [130]. It was reported that oncolytic HSV-1 mutant R3616 was adsorbed by antigen-specific lymphocytes of *mice* with anti-tumor immunity, and its cure rate was approximately 40% in the treatment of colon cancer *mice* models [131]. As mentioned above, cellular delivery of viruses, which can avoid viral attacks by immune cells, is a viable option for achieving multiple OV administrations. Cell-delivered OVs can improve the virus targeting to tumor cells and virus stability in vivo. However, most of the current research includes the use of cell-delivered OVs for the treatment of solid tumors in the early stage [132].

### 5.2. Enhancing the Targeting of Tumors

Besides the above-mentioned strategies to overcome immune clearance, enhancing tumor targeting can also effectively improve the efficacy of intravenous OV. Therefore, to minimize the non-targeted toxic effects on the organism from the prolonged peripheral circulation, the virus must be highly selective for the tumor site and have a high local colonization rate and quantity in the tumor. Tumor-specific promoters, for example, can enhance OV enrichment at tumor sites, or tumor-targeting antibodies and peptides can be expressed on the virus surface to enhance the targeting of viruses [133,134,135]. It was discovered that intravenous administration of recombinant OV HSV1716EGFR, which can express anti-epidermal growth factor receptor (EGFR) scFv, increased the accumulation of virus in tumor tissue by 8-fold 24 days following injection. However, the unmodified virus was barely detectable at the tumor site and was not detected in normal tissue [136]. This indicates that intravenous injection of tumor OV is safe and provides a solid basis for clinical trials. This targeting strategy typically involves genetic modification of HSV-1 OV, such as knocking out the viral *TK* and *ICP34.5* genes to improve the tumor-killing ability or modifying the promoter of ICP0 into a tumor-specific promoter [e.g., human telomerase reverse transcriptase, hypoxia-responsive, carcinoembryonic antigen CEA, prostate-specific antigen, alpha-fetoprotein, alpha-lactalbumin, and mucin1 promoters (*DF3/MUC1*)], enabling their specific replication in tumor cells [137,138].

## 6. Conclusions

Tumor immunotherapy, represented by the immune checkpoint inhibitor PD-1 antibody and CAR-T cell immunotherapy, has significantly increased the overall survival of patients with advanced cancer in recent years [139,140]. However, its therapeutic efficacy in most cold tumors and PD-L1-negative solid tumors remains limited. Therefore, alternative therapeutic strategies are urgently required [141]. HSV-1 oncolytic virus, as an excellent gene vector, can not only effectively lyse tumor cells but also improve the TME by carrying various cytokines, which bodes well for tumor immunotherapy. Currently, most of the clinical studies on HSV-1 oncolytic virus have been conducted via intratumoral injection; however, since intratumoral injection requires complex surgical procedures and there are barriers to delivering OVs to distant metastases, it is crucial to use injection methods according to the tumor type and location to enhance the efficacy of HSV-1 oncolytic virus [30].

Intravenous administration is occasionally a favorable option for the treatment of metastatic tumors. Although systemic administration of OV has been reported to be safe, there has been no breakthrough in efficacy. This is mainly owing to the presence of antiviral immunity that allows rapid clearance of these drugs, such as the possibility of HSV-1 OV in the blood being cleared by the body’s immune system before it reaches the tumor tissue [142]. Therefore, when using intravenous OV, the balance between antiviral immunity in the blood and immune activation in the tumor should be considered to ensure safety and minimize the clearance of intravenous OV by the body to ensure that sufficient doses of OV reach the tumor site to function.

As previously mentioned, the problems of immune clearance and poor targeting that exist when intravenous OV are administered have been largely addressed, and numerous strategies are being optimized and refined as scientific research advances. The use of immunosuppressants reduces the clearance of OV by the body; however, whether they suppress anti-tumor immunity and affect the tumor immune microenvironment remains to be investigated. Therefore, when using immunosuppressants to overcome the immune clearance problem of intravenous administration of OV, the relationship between antiviral and anti-tumor immunity needs to be considered, and investigating the differences between the two and identifying inhibitors or target molecules that can specifically inhibit antiviral immunity are the main directions for the future development of this strategy. Furthermore, to overcome immune clearance by systemic intravenous administration of OVs using cellular vectors, the primary focus of future research will be to explore the interaction between the cellular vector and the tumor site and to investigate how to ensure the large-scale expansion of the OV after the cellular vector reaches the tumor site. Combining immune checkpoint inhibitors with nanomaterials encapsulated with OVs for systemic delivery to address the immune clearance and targeting issues of intravenous injection has produced positive research results in tumor *mice* models [143], and the combination of multiple strategies will be the trend to achieve effective systemic delivery of OVs.

Delayed tumor growth and even tumor regression at metastatic lesions have been observed in animal and human models following local administration of OVs. Meanwhile, the adaptability of genetic engineering has caused a renaissance in the field of HSV-1 OV genetic modification. Therefore, continuous optimization of the viral backbone to improve the local tumor lysis and immune activation ability while maintaining safety remains an important direction for the development of the HSV-1 OV. In addition, although it has certain efficacy in *mice* models, due to the genetic differences between humans and *mice*, it remains to be proven whether these improved strategies will be equally effective in humans.

Conclusively, many preclinical trials have demonstrated significant anti-tumor effects of HSV-1 oncolytic virus on advanced cancer treatment with a high safety profile. Although some studies have revealed the basic mechanisms of immune activation and immune escape by HSV-1 oncolytic virus, for intravenous administration of HSV-1 oncolytic virus, a comprehensive understanding of the new mechanisms of host, virus, and immune interactions and the development of new intravenous therapeutic strategies may produce more effective cancer treatments soon.

## Figures and Tables

**Figure 1 ijms-24-03681-f001:**
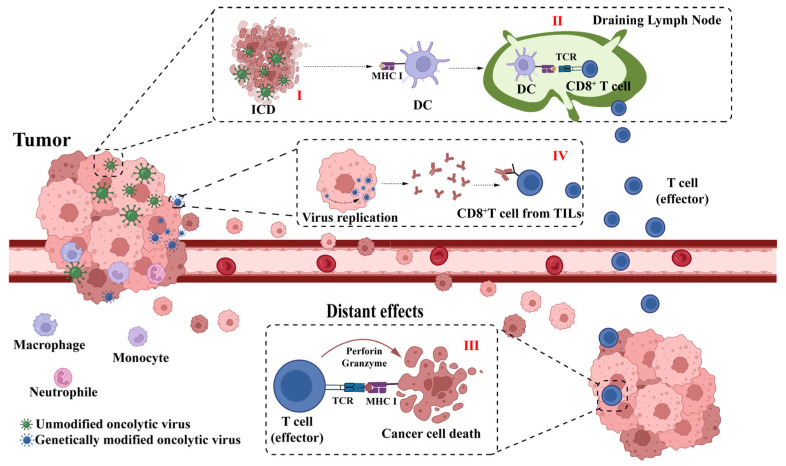
Mechanism of action of oncolytic virus therapy. I: Oncolytic virus (OV) can directly lyse tumor cells and induce immunogenic death of tumor cells, thus, releasing tumor antigens. II and III: Tumor antigens are presented by activated antigen-presenting cells (APCs), which can activate CD8^+^ T cells in the draining lymph nodes, thus, inducing an organism-specific anti-tumor immune response and playing a role in killing distant metastases. IV: Genetically modified OV promotes antitumor immunity. Genetically engineered OV carrying exogenous genes can improve the immunosuppressed tumor microenvironment (TME) by simultaneously expressing exogenous genes during replication, relieving T-cell suppression, and promoting T-cell activation.

**Figure 2 ijms-24-03681-f002:**
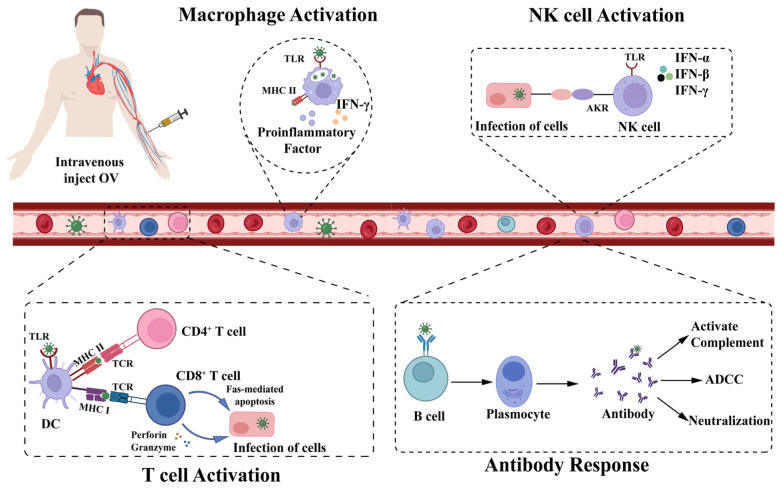
Immune recognition effect induced by intravenous injection of oncolytic virus (OV). When the OV is administered intravenously, the cells of the body’s immune system develop antiviral immunity. Dendritic cells (DC) present viral antigens to T cells and activate them. CD8^+^ T cells can play a direct toxic role in infected cells by releasing perforin and granzyme and can also cause apoptosis through the Fas pathway; B cells produce neutralizing antibodies by recognizing viral antigens to prevent re-infection of cells, and these antibodies can mediate antibody-dependent cell-mediated cytotoxic effects or activate the complement pathway to produce antiviral effects; macrophages produce large amounts of pro-inflammatory factors during the early stages of infection and phagocytose the virus through cytokinesis; after the downregulation of major histocompatibility complex (MHC) class I molecules on the surface of infected cells, natural killer (NK) cells can recognize the signal to directly kill cells, and infected cells can also activate the receptor through NK cells (AKR activates NK cells to produce pro-inflammatory factors to play a defensive role against viral infections).

**Figure 3 ijms-24-03681-f003:**
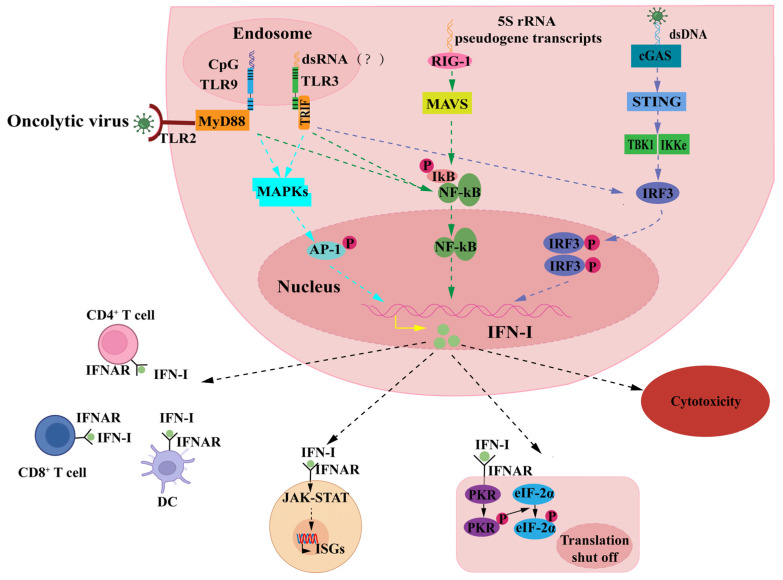
Antiviral effect of IFN-I. When viruses infect cells, they can either activate mitogen-activated protein kinase (MAPK) phosphorylation in the nucleus or promote nuclear factor kappa B (NF-κB) dissociation from IκB via myeloid differentiation primary response protein 88 (MyD88) signaling after binding pathogen-associated molecular pattern (PAPMs) to Toll-like receptors 2 (TLR2). Herpes simplex virus type 1 (HSV-1) contains genomes rich in CpG DNA motifs and was demonstrated to activate type I interferon (IFN-I) secretion via TLR9 [75,76]. TLR3 can interact with double-stranded HSV-1, a non-transcribed RNA intermediate, during infection through a MyD88-independent pathway resulting in the activation of NF-κB and interferon regulatory factor (IRF)-3 [77]. In addition, retinoic acid-inducible gene l-like receptors (RIG-I) recognize host 5S rRNA pseudogene transcripts but not HSV-1 genomic-derived RNA after activating IFN-I expression via the NF-κB signaling pathway [78], and viral dsDNA interacts with cyclic GMP-AMP synthase (cGAS) via the cGAS-STING signaling pathway to initiate IFN-I expression. IFN-I binds to IFN receptor and induces more than 200 IFN-stimulated genes (*ISGs*) via Janus kinase/signal transducers and activators of transcription (JAK-STAT) pathway to produce comprehensive antiviral effects, and it activates relevant immune cells to promote antiviral immunity. This causes cellular transcriptional arrest through the activation of the protein kinase R (PKR) signaling pathway and exerts direct toxic effects on infected cells to resist viral infection.

**Figure 4 ijms-24-03681-f004:**
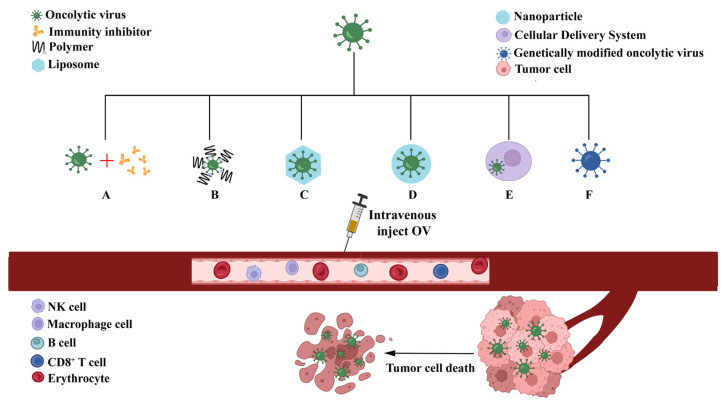
Strategies to overcome immune clearance and enhance targeting of tumors. These strategies can be applied to prevent the clearance of oncolytic virus (OV) by the body’s antiviral immunity and to improve the targeting of OV when systemic administration of intravenous OV is achieved for the treatment of tumors. (**A**): Combined with immune inhibitor. (**B**): Modification of OV using polymers. (**C**): Use of liposomes to encapsulate OV. (**D**): Use of nanoparticles to encapsulate OV. (**E**): Delivery of OV using cell delivery vectors. (**F**): Genetic editing of OV to improve targeting of tumors.

## Data Availability

All the data can be found in the PubMed database-https://pubmed.ncbi.nlm.nih.gov/ (accessed on 3 January 2023) or under the links cited of cited websites.

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
