# Peer review of "The Dilemma of HSV-1 Oncolytic Virus Delivery: The Method Choice and Hurdles"

_ijms, 2023, doi:10.3390/ijms24043681_

Round 1

Reviewer 1 Report

This manuscript describes specifically current difficulties in the intravenous administration of oncolytic HSV1. However, the contents of the information are limited and mainly describe general basic knowledge of immune responses after oncolytic virus administration. If the authors hope to focus on the trial of overcoming IV method in oHSV1, they should describe in detail each strategy how to evade immune responses and the interpretation of results in the papers. I would recommend this manuscript to publish after the major revision of its contents.   

Author Response

Point 1: This manuscript describes specifically current difficulties in the intravenous administration of oncolytic HSV1. However, the contents of the information are limited and mainly describe general basic knowledge of immune responses after oncolytic virus administration. If the authors hope to focus on the trial of overcoming IV method in oHSV1, they should describe in detail each strategy how to evade immune responses and the interpretation of results in the papers. I would recommend this manuscript to publish after the major revision of its contents.

Response 1: We thank you very much for giving us an opportunity to revise our manuscript. Upon your constructive suggestion, we made revision accordingly. In the section 5 of the revised manuscript (Pages 8-10 of 21, lines 352-439), we describe in detail how each strategy evades the immune response and add related studies to support it. We also present our perspectives on the future development of intravenous administration (Page 8 of 21, lines 347-351. Page 11 of 21, lines 453-488). The changes have been marked by red in the revised manuscript. Once again thank you very much for your comments and suggestions.

Reviewer 2 Report

The authors have reviewed the topic of delivery of oncolytic herpes viruses, focusing on the potential hurdle of intravenous delivery.

The authors have done the summary of literature quite well, yet there are not enough of authors’ own critical evaluations and perspectives on the future developments. We need to hear more for that in order for this review to be useful for readers of this high-quality journal.

Minor points:

1.      Page 6 of 18, lines 240-241. The phrase “oncolytic bovine poxvirus JX594” is not correct. JX-594 (now called Pexa-Vec) is a vaccinia virus, not a bovine poxvirus. I know that even Encyclopedia Britannica made the same mistake (by saying vaccinia virus is cowpox virus). Although the exact origins of vaccinia virus are uncertain, vaccinia virus may represent a hybrid of the variola and cowpox viruses.

2.      Section 4 (potential HSV-1 immune clearance problems in intravenous administration)

For the whole section, the authors just cited just one reference (ref #58). Are there any other better studies on this issue? If there are, please discuss the findings in these studies.

3.      Section 5 (page 8), line 347. “Autoimmune clearance” may not be right phrase. “autoimmune” refers to immune attack to its own cells or tissue, not the virus that is foreign.

4.      The following existing references may need correct or additional information:

Ref #5. No page numbers. Is it an abstract?

Ref #50. Name of the journal?  (Mol Ther Oncolytics)

Ref #56. No page numbers. Is it an abstract?

Ref #113. Missing the article number.

Author Response

Point 1: The authors have reviewed the topic of delivery of oncolytic herpes viruses, focusing on the potential hurdle of intravenous delivery.

The authors have done the summary of literature quite well, yet there are not enough of authors’ own critical evaluations and perspectives on the future developments. We need to hear more for that in order for this review to be useful for readers of this high-quality journal.

Response 1:  We very much appreciate your constructive suggestions and comments. Upon your suggestion, we made revision accordingly. In section 5 and 6, we present our perspectives on the future development of intravenous administration (Page 8 of 21, lines 347-351. Page 11 of 21, lines 453-488). We also invited native English speaker for language corrections. Following are the answers to your questions point to point. The changes have been marked by red in the revised manuscript and indicated in the following.

Point 2: Page 6 of 18, lines 240-241. The phrase “oncolytic bovine poxvirus JX594” is not correct. JX-594 (now called Pexa-Vec) is a vaccinia virus, not a bovine poxvirus. I know that even Encyclopedia Britannica made the same mistake (by saying vaccinia virus is cowpox virus). Although the exact origins of vaccinia virus are uncertain, vaccinia virus may represent a hybrid of the variola and cowpox viruses.

Response2: Thanks for your careful check and correction. We have corrected it in the revised manuscript (Page 6 of 21, lines 241-242).

Point 3: Section 4 (potential HSV-1 immune clearance problems in intravenous administration)

For the whole section, the authors just cited just one reference (ref #58). Are there any other better studies on this issue? If there are, please discuss the findings in these studies.

Response 3: We thank you for your valuable suggestions and have made additions according to your suggestions (Page 6 of 21, lines 257-263).

Point 4:Section 5 (page 8), line 347. “Autoimmune clearance” may not be right phrase. “autoimmune” refers to immune attack to its own cells or tissue, not the virus that is foreign.

Response 4: Thanks for your careful check and correction. We have corrected it in the revised manuscript (Page 9 of 21, line 360).

Point 5: The following existing references may need correct or additional information:

Ref #5. No page numbers. Is it an abstract?

Response : Yes. This is an abstract. Can click the following URL to find out: https://ascopubs.org/doi/abs/10.1200/JCO.2020.38.4_suppl.117. (Page 13 of 21, line 530)

Ref #50. Name of the journal?  (Mol Ther Oncolytics)

Response :  The name of the journal is Molecular Therapy. (Page 16 of 21, line 645)

Ref #56. No page numbers. Is it an abstract?

Response : Yes. This is an abstract. Can click the following URL to find out: https://aacrjournals.org/cancerres/article/81/13_Supplement/2597/668346/Abstract-2597-Non-clinical-studies-of-systemic. The page numbers are 2597. (Page 16 of 21, line 663)

Ref #113. Missing the article number.

Response : Thanks for your careful check and correction. We have corrected it in the revised manuscript. (Page 20 of 21, line 844, Ref #127.)

The above contents are our responses to your valuable questions or suggestions. Thank you very much for your patience! 

Round 2

Reviewer 1 Report

The manuscript improved very well. Especially, the description of the previous and current strategies in detail for immune evasion is attractive to readers in this field. So I could recommend this for publishing as it is.

Reviewer 2 Report

The authors have made significant improvements in those areas this reviewer had discussed for revisions on the original version of the manuscript.

There are a few minor errors or misses in these references:

1.     Ref #10. Barker DD and Berk A. J. J.V. The second author’s name is “Berk A.J.”, not “Berk A.J. J. V.”.

2.     Ref #41. The name of the journal? Also please check the error in last authors’ initials: “Tanabe, K.K.J.T.J.o.c.i.”?

3.     Ref #51. The name of the journal?

4.     Ref #117. The last three authors have been missing from the list. Please follow the format of the journal to make changes (either listing all of them, or using “et al.” according to the required format)”

5.     Ref #136. The name of the journal?